# Investigation of the Morphological Structure of Needle-Free Electrospun Magnetic Nanofiber Mats

**Al Mamun** [1], **Michaela Klöcker** [2], **Tomasz Blachowicz** [3] and **Lilia Sabantina** [1,*]

1 Junior Research Group "Nanomaterials", Faculty of Engineering and Mathematics, Bielefeld University of Applied Sciences, 33619 Bielefeld, Germany; al.mamun@fh-bielefeld.de
2 Faculty of Engineering and Mathematics, Bielefeld University of Applied Sciences, 33619 Bielefeld, Germany; michaela.kloecker@fh-bielefeld.de
3 Institute of Physics—Center for Science and Education, Silesian University of Technology, 44-100 Gliwice, Poland; tomasz.blachowicz@polsl.pl
* Correspondence: lilia.sabantina@fh-bielefeld.de

**Abstract:** Electrospun magnetic nanofibers are promising for a variety of applications in biomedicine, energy storage, filtration or spintronics. The surface morphology of nanofiber mats plays an important role for defined application areas. In addition, the distribution of magnetic particles in nanofibers exerts an influence on the final properties of nanofiber mats. A simple method for the production of magnetic nanofiber mats by the addition of magnetic nanoparticles in an electrospinning polymer solution was used in this study. In this work, magnetic nanofibers (MNFs) were prepared by needle-free electrospinning technique from poly(acrylonitrile) (PAN) in the low-toxic solvent dimethylsulfoxide (DMSO) and 20 wt% $Fe_3O_4$ at different parameter conditions such as PAN concentration, voltage and ultrasonic bath. The distribution of nanoparticles in the fiber matrix was investigated as well as the chemical and morphological properties of the resulting magnetic nanofibers. In addition, the surface morphology of magnetic nanofiber mats was studied by confocal laser scanning microscope (CLSM), scanning electron microscope (SEM), Fourier transform infrared microscope (FTIR) and ImageJ software, and distribution of $Fe_3O_4$ particles in the matrix was investigated by energy dispersive X-ray spectroscopy (EDX).

**Keywords:** needleless electrospun nanofiber mats; magnetic nanoparticles; energy dispersive X-ray spectroscopy (EDX); scanning electron microscopy (SEM)



## 1. Introduction

Electrospinning is the process of producing polymer nanofibers through electrostatic forces, the source of which is a high voltage [1]. The production of nanofibers and mats by electrospinning technology allows the easy production of nanofiber mats from bio-based, man-made polymers or polymer blends and by adding particles of ceramics, metals or metal oxides etc. [2–5]. New materials are promising for use in filtration, energy, biomedicine, electromagnetic shielding, neuromorphic computing, spintronics or energy storage [6–10]. Nanofibers with magnetic properties (MNFs) produced with electrospinning technology are currently attracting great interest from both academia and industry due to the development of new materials with magnetic and conducting properties [11].

In addition, the energy required for electrospinning technology in small production series is very low compared to the use of conventional technologies, which also has a significant impact on the cost benefits and environmental friendliness of production [12].

Generally speaking, a distinction can be made between needle-based and needle-free electrospinning technologies [13–15]. The needle-free technique offers the production of nanofiber mats on an industrial scale, for example, with the "Nanospider™" electrospinning machine from Elmarco, Czech Republic. Nanofibers are created in a high-voltage field, and electrospinning machines include upper and lower wires, a carriage with a polymer

solution, and a nozzle through which polymer solution exits and coats the lower wire. Thus, multiple tailor cones are generated simultaneously along the coated wire, producing nanofibers that are deposited on a nonwoven substrate (e.g., polypropylene) below the upper wire [16,17].

The needle-based electrospinning machine consists of a pump, syringe, high-voltage power supply, needle and collector. The positive pole of the power supply is connected to the needle and the negative pole to the collector. The polymer solution is introduced into the syringe and pressed out in a controlled manner; simultaneously, Taylor cones are formed and nanofibers are deposited [18].

Needle-based (1a) and needle-free (1b) techniques are shown in Figure 1. With both electrospinning techniques, it is possible to produce magnetic nanofiber mats (MNFs) by simply adding magnetic particles to the electrospinning solution. The schematic representation of the production of nanofibers with particles is shown in Figure 1c.

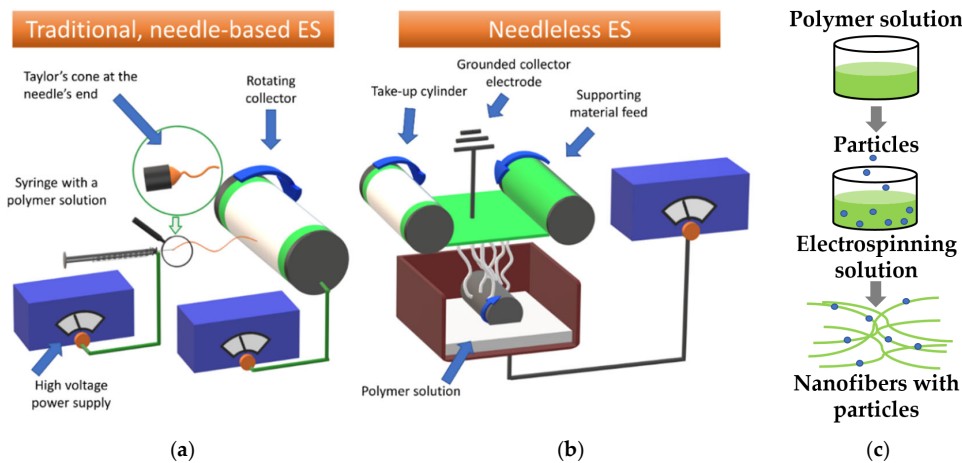

**Figure 1.** (**a**) Needle-based and (**b**) needleless electrospinning (ES) techniques. Reprinted from study [19], originally published under a CC-BY 4.0 license; (**c**) Manufacturing process of nanofibers with particles.

The magnetic properties of nanofibers and nanofiber composites are of great interest and the morphology has an influence on the mechanical and chemical properties of the final nanofiber mats. The strategies for an optimal distribution of magnetic nanoparticles are discussed in the literature and various approaches are proposed, such as coating of particles or ultrasonication [20–22]. The study by Lalatonne et al. notes that conventional mixing methods contribute to particles connecting and aggregating during the formation of the materials due to magnetic dipole and van der Waals interactions [20]. In the study by Tanaka et al., surface treatment of magnetite with oleic acid or stearic acid was performed and ultrasonic treatment was used to study the influence of magnetite dispersion on the tensile properties of magnetite/PLA nanofiber nonwoven fabrics. It was found that agglomerations of magnetic particles negatively affect the mechanical properties [21]. In the study by Chowdhury et al., magnetically responsive, mechanically stable and highly flexible piezoelectric composite fiber mats made of magnetite polyvinylidene fluoride ($Fe_3O_4$-PVDF) were produced in the single-stage electrospinning process and ultrasonication treatment for better dispersion was used [22].

The small particles have a tendency to agglomerate and, most of the time, beads are undesirable and different strategies such as ultrasonic bath are used for this purpose. For some applications, the beads are desired, such as when electrospun magnetic nanofiber mats are envisioned as new hardware for neuromorphic computing. In this case, the beads will be designated for data storage and magnetically doped polymer nanofibers for data transmission [23,24].

Our previous studies have shown that poly(acrylonitrile) (PAN) nanofiber mats with magnetite (20 wt%) or nickel ferrite nanoparticles (25 wt% in the spinning solution) electro-

spun by the needleless machine "Nanospider Lab", as in this study, showed beads in which the magnetic nanoparticles agglomerated [24,25]. As in the previous study, mechanical separation by ultrasonic treatment was used in our material system for preparation of the electrospinning solution [5,26].

In this study, magnetic nanofiber mats were prepared from poly(acrylonitrile) (PAN) dissolved in low-toxic solvent dimethyl sulfoxide (DMSO) and by adding magnetite ($Fe_3O_4$) nanoparticles using needleless electrospinning technique. With change of spinning parameters and the use of ultrasonic bath, effects on the morphological changes and fiber distribution were observed. The defined distribution of magnetic particles in nanofiber mats poses a great challenge for special applications and, according to research, there are very few papers on this approach, which underlines the novelty of this topic, to which our study also contributes.

## 2. Materials and Methods

The Nanospider Lab (Elmarco, Liberec, Czech Republic) needleless electrospinning machine was used for the production of nanofiber mats. Nanofibers were collected on a substrate of polypropylene (PP) and then separated from PP. The following electrospinning parameters were used: high voltages of 65 kV and 80 kV, nozzle diameter of 0.9 mm, carriage speed of 150 mm/s, positive electrode-to-substrate distance of 240 mm, and a ground electrode-to-substrate distance of 50 mm. The temperature in the chamber of the electrospinning machine was 23 °C and relative humidity was set to 32–33%. The electrospinning duration was determined to 20 min.

The electrospinning solutions were prepared according to the following procedure. Firstly, 14 wt%, 16 wt% and 18 wt% of PAN (X-PAN, Dralon, Dormagen, Germany) were dissolved in dimethyl sulfoxide (DMSO, min 99.9%, S3 Chemicals, Bad Oeynhausen, Germany). Then, the polymer solutions were mixed with magnetic stirrer at room temperature for 1 h. Afterwards, 20 wt% magnetic particles $Fe_3O_4$ (magnetite, particle size 50–100 nm, Merck KGaA, Darmstadt, Germany) were added to the PAN solutions and stirred manually first. The amount of magnetite in the total spinning solutions is thus 16.7 wt%. Then, some samples were treated for 40 min at 35 °C and a frequency of 37 kHz in an ultrasonic bath (Elmasonic Select, Elma Schmidbauer GmbH, Singen, Germany).

These magnetic particles were chosen because their magnetic properties are already known from previous studies of our group. Former simulations of the magnetic properties of PAN/magnetite nanofibers revealed coercive fields of approx. 240 Oe for 14.5 vol.% magnetite in the nonmagnetic matrix of the fiber [27]. On the other hand, former experimental investigations resulted in coercive fields of approx. 200 Oe for 28.6 vol.% magnetite in the nanofiber mat [28]. Due to the small difference in the coercive field upon doubling the magnetite concentration in the nanofiber mat, the recent samples can be expected to show coercive fields in the range of 200–240 Oe.

The magnetic properties of PAN/magnetite nanofiber mats were investigated in our previous study by Fokin et al. [28]. It was found that the magnetic properties of nanofiber mats containing magnetite compared to raw nanofiber mats were dependent on the nanoparticle diameters and potential thermal post-treatment. In addition, the micromagnetic simulations revealed the properties of the magnetic materials [28].

Nanofiber diameters were determined based on analysis of SEM micrographs and using ImageJ (software version 1.53e, 2021, National Institutes of Health, Bethesda, MD, USA). Nanofiber diameter distribution was calculated by measuring 100 fibers. Optical studies were performed using confocal laser scanning microscope (CLSM), VK-8710 (Keyence, Osaka, Japan).

More detailed studies of surface morphology were performed with Sigma 300 VP (Carl Zeiss Microscopy GmbH, Oberkochen, Germany) scanning electron microscope (SEM) and energy dispersive X-ray spectroscopy (EDX) with an Oxford EDX system equipped with an 80 mm$^2$ X-Max-detector using AZtec 3.1 (Oxford Instruments PLC, Abingdon, UK).

For Fourier transform infrared (FTIR) spectroscopy, an Excalibur 3100 (Varian, Inc., Palo Alto, CA, USA) with a spectral range of 4000 cm$^{-1}$ to 700 cm$^{-1}$ was used. In each case, 32 scans were averaged as well as atmospheric noise corrected. The overview of the samples is shown in Table 1.

**Table 1.** Sample overview.

| | |
|---|---|
| 14 wt% PAN | |
| 16 wt% PAN | High voltage of 65 kV and ultrasonic bath |
| 18 wt% PAN | |
| 14 wt% PAN | |
| 16 wt% PAN | High voltage of 80 kV and ultrasonic bath |
| 18 wt% PAN | |
| 14 wt% PAN | |
| 16 wt% PAN | High voltage of 80 kV and without ultrasonic bath |
| 18 wt% PAN | |

## 3. Results and Discussion

The morphologies of the studied nanofiber mats are shown in the CLSM micrographs in Figures 2–4. When observing the surface morphology of the nanofiber mats with 14 wt%. PAN, it was found that all nanofiber mats show relatively straight nanofibers with some beads and membrane sites, and no visible differences are observed between the nanofiber mats prepared with different high voltage of 65 kV and 80 kV with or without the employment of ultrasonic bath (cf. Figure 2a–c).

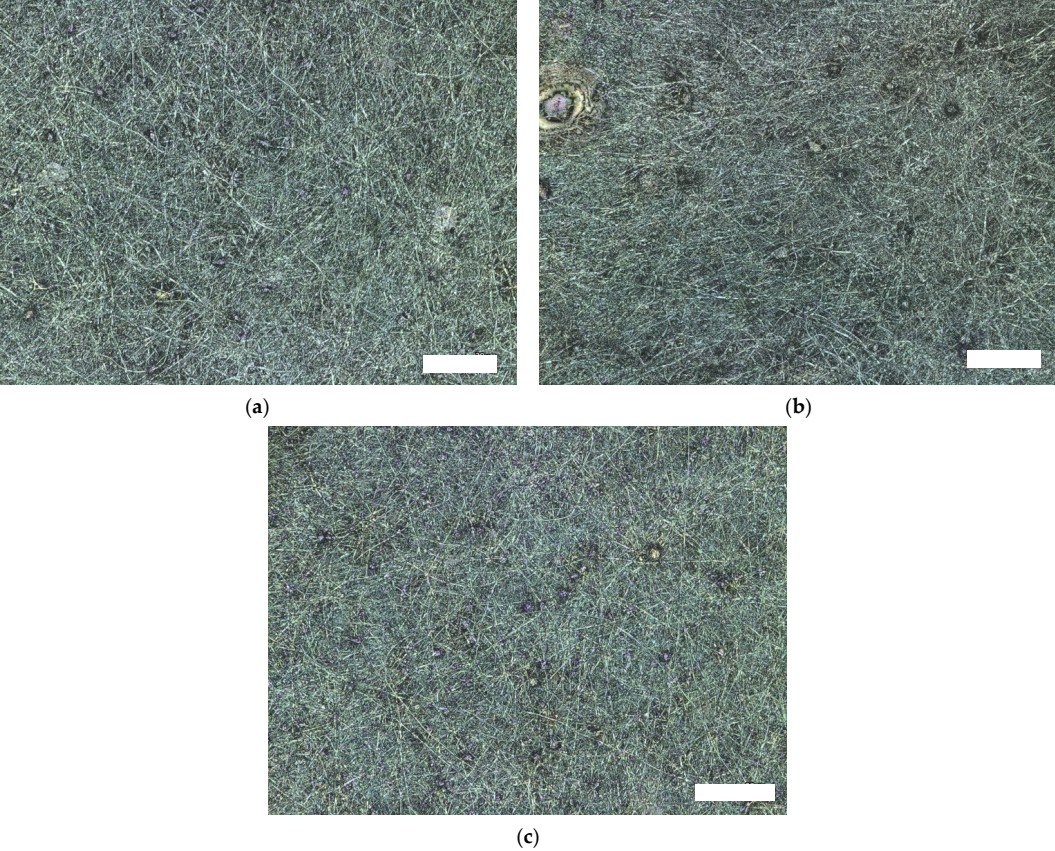

(a)                                                                    (b)

(c)

**Figure 2.** Confocal laser scanning microscope (CLSM) images of the magnetic nanofiber mats: (**a**) 14 wt% PAN with 65 kV voltage and 40 min ultrasonic bath; (**b**) 14 wt% PAN with 80 kV voltage and 40 min ultrasonic bath; (**c**) 14 wt% PAN and 80 kV voltage without ultrasonic bath. Scale bars indicate 20 μm.

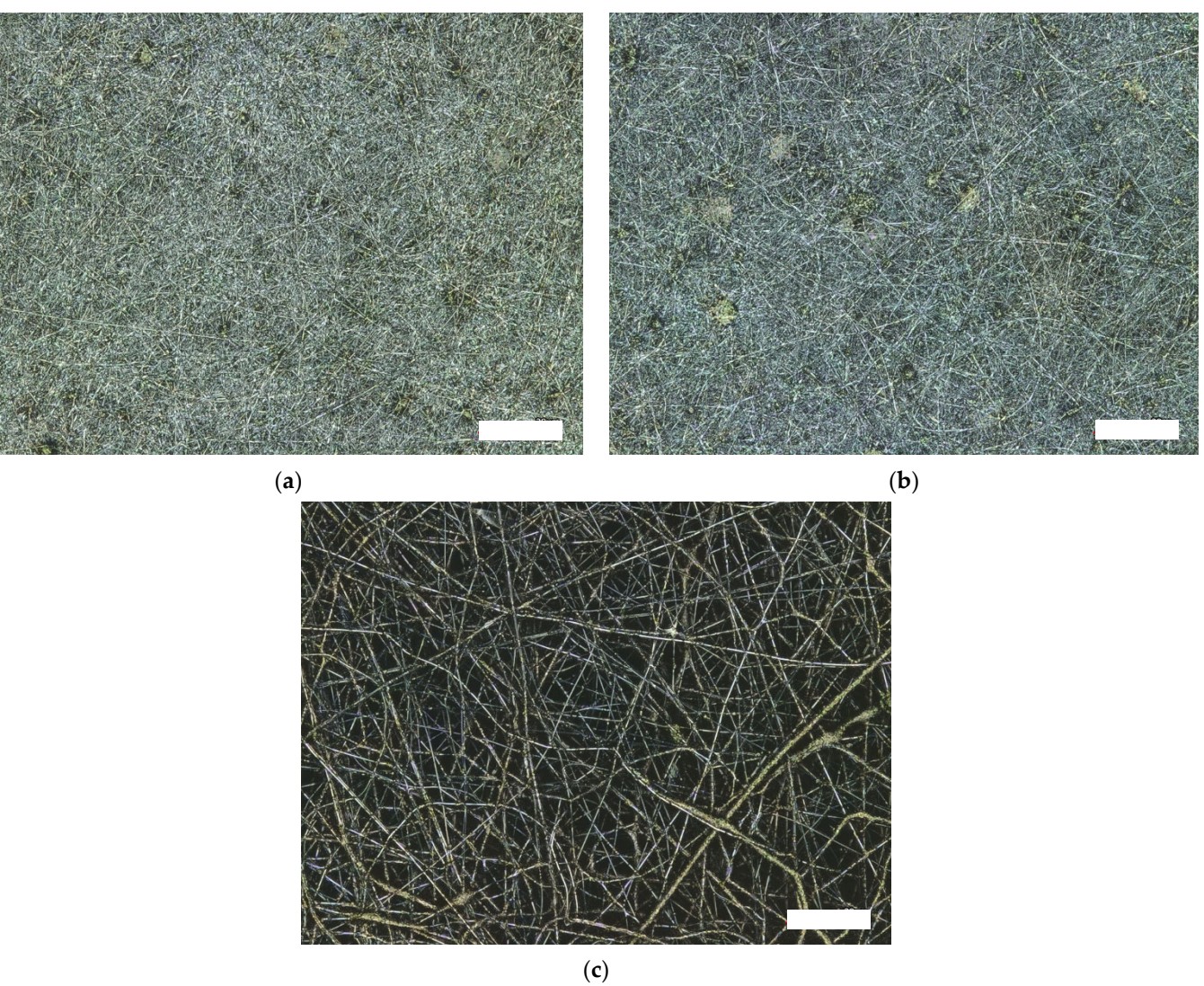

**Figure 3.** Confocal laser scanning microscope (CLSM) images of the magnetic nanofiber mats: (**a**) 16 wt% PAN with 65 kV voltage and 40 min ultrasonic bath; (**b**) 16 wt% PAN with 80 kV voltage and 40 min ultrasonic bath; (**c**) 16 wt% PAN and 80 kV voltage without ultrasonic bath. Scale bars indicate 20 μm.

When observing the surface morphology of the magnetic nanofiber mats with 16 wt% PAN, it can be seen that the nanofiber mats with 65 kV (Figure 3a) and 80 kV (Figure 3b) still show a few visible differences in morphology. Nanofiber mats prepared with 16 wt% PAN contain straight fibers with small beads and membrane areas (Figure 3a,b). In this case, membrane areas mean the connected molten areas, which can be seen as light spots on the CLSM images (see Figure 3a,b). It can happen that the solvent does not completely evaporate and, therefore, the nanofibers are melted together and form membrane areas. In our previous study, by varying the distance between the electrodes, different morphologies of the nanofiber mats were fabricated, allowing the desired fiber-to-membrane ratio by varying this parameter [29].

The nanofiber mat produced at 80 kV voltage shows more membrane areas, but this could be due to the reason that the images show only a small part of the total sample and are therefore not always reproducible, as was found in the study by Wortmann et al. [27]. However, when looking at nanofiber mats produced at 80 kV without ultrasonic bath (Figure 3c), a clear difference is visible. The individual MNFs look significantly thicker (Figure 3c) compared to nanofiber mats in Figure 3a,b.

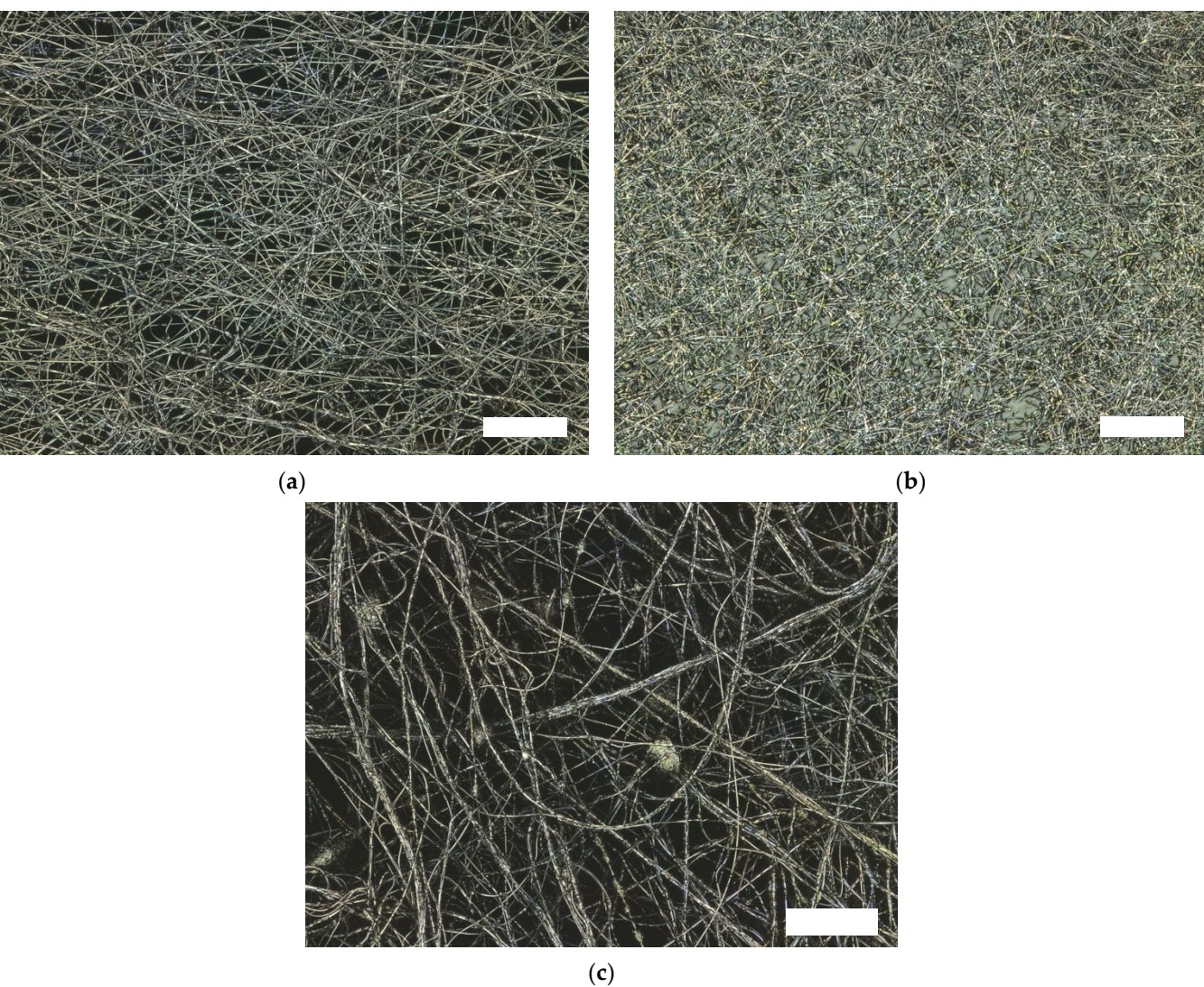

**Figure 4.** Confocal laser scanning microscope (CLSM) images of the magnetic nanofiber mats: (**a**) 18 wt% PAN with 65 kV voltage and 40 min ultrasonic bath; (**b**) 18 wt% PAN with 80 kV voltage and 40 min ultrasonic bath; (**c**) 18 wt% PAN and 80 kV voltage without ultrasonic bath. Scale bars indicate 20 µm.

At 18 wt% PAN, MNFs electrospun at 80 kV (Figure 4a) and 65 kV (Figure 4b) voltage look much thicker compared to 14 wt% (see Figure 2a,b) and 16 wt% PAN (see Figure 3a,b) nanofiber mats. It is known from previous studies that with the increase of polymer concentration in electrospinning solution or upon adding of particles, the nanofiber diameters increase [30–32]. It is also visible that without the use of ultrasonic bath (Figure 4c), the MNFs' diameter is much larger and the beads and agglomerations are clearly visible.

CLSM images provided the first insight and SEM micrographs were used for a deeper investigation (see Figure 5). Here, nanofiber mats with 16 wt% PAN are discussed in more detail because according to our previous studies, 16% PAN in the electrospinning solution and 80 kV voltage were used as the best electrospinning parameters for producing uniform, well-separated nanofibers [5,16,17,26]. The morphology of the MNFs is shown in Figure 5 by SEM micrographs at 5000× magnification. The nanofiber mats manufactured at 65 kV (Figure 5a) and 80 kV (Figure 5b) voltage using an ultrasonic bath show relatively straight fibers with some beads as well as membrane areas. The difference of surface morphology of nanofiber mats between 65 kV (Figure 5a) and 80 kV (Figure 5b) voltage is not that large. The MNFs produced at 65 kV voltage show some spots with agglomerations, and at 80 kV

voltage, fewer agglomerations are visible. The MNFs fabricated at 80 kV voltage without using an ultrasonic bath (see Figure 5c) show a different nanofiber morphology with thicker nanofibers, interfiber bonding, and denser beads and more agglomeration spots. The effect of interfiber bonding and denser beads in Figure 5c will be investigated in further study; whether settings will lead to the same nanofiber morphology cannot be answered with this study yet.

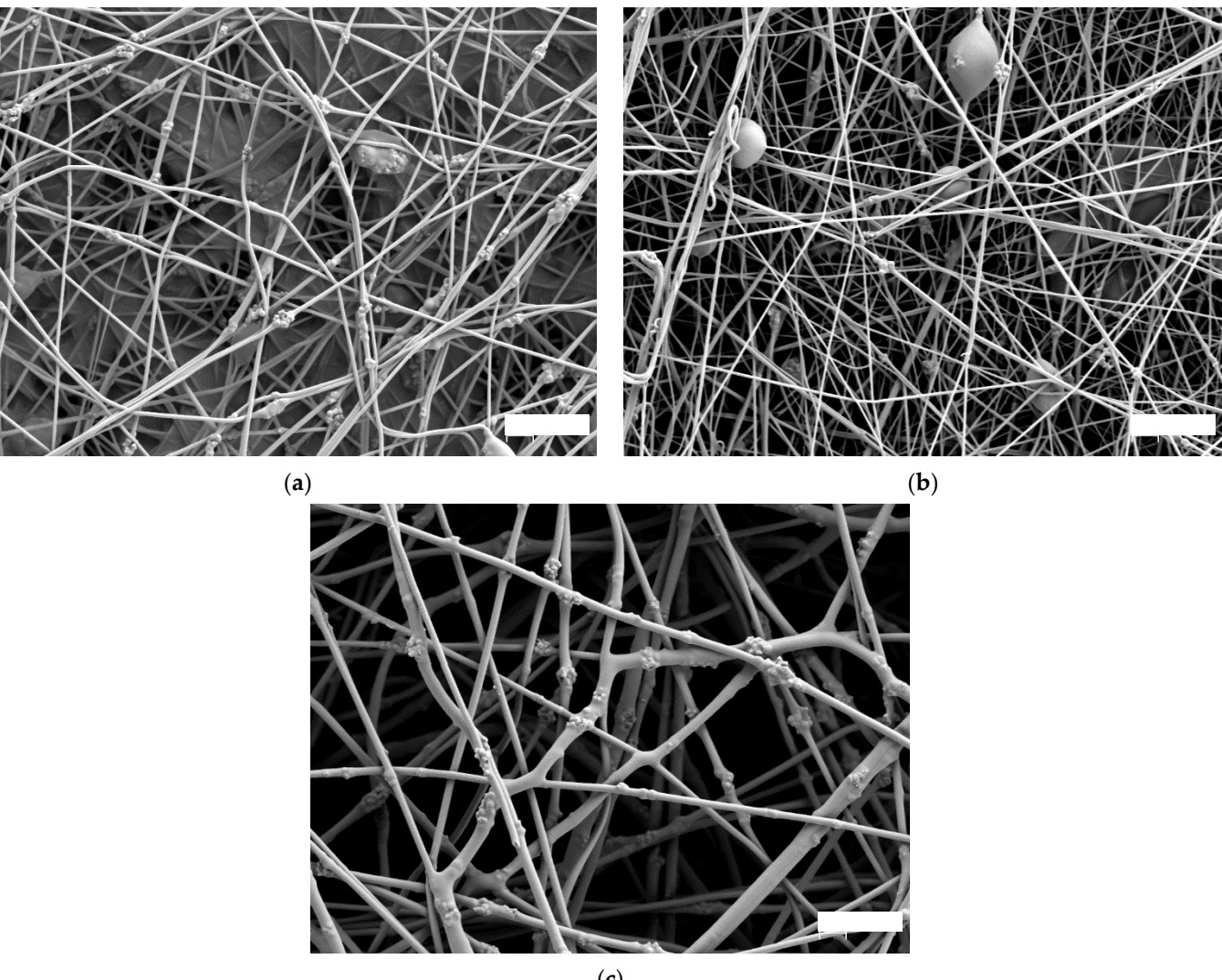

**Figure 5.** Scanning electron microscope (SEM) micrographs of the magnetic nanofiber mats: (**a**) 16 wt% PAN with 65 kV voltage and 40 min ultrasonic bath; (**b**) 16 wt% PAN with 80 kV voltage and 40 min ultrasonic bath; (**c**) 16 wt% PAN and 80 kV voltage without ultrasonic bath. Scale bars indicate 3 μm.

It is also important to note that the individual MNFs (see Figure 5c) are bound to each other and form a network in contrast to the MNFs in Figure 6a,b, where they are separated from each other and do not form networks.

Furthermore, the MNFs' diameter distribution appears to be much broader compared to the Figure 5a,b nanofiber mats.

The magnetic properties of most nanofibers depend more on the distribution of the nanoparticles than on the total amount for production of nanofibers for relevant applications [33,34]. In addition, it can be seen in Figure 6 that the SEM micrographs confirm the aggregation of magnetic nanoparticles and show large nodules with magnetic clusters in all samples.

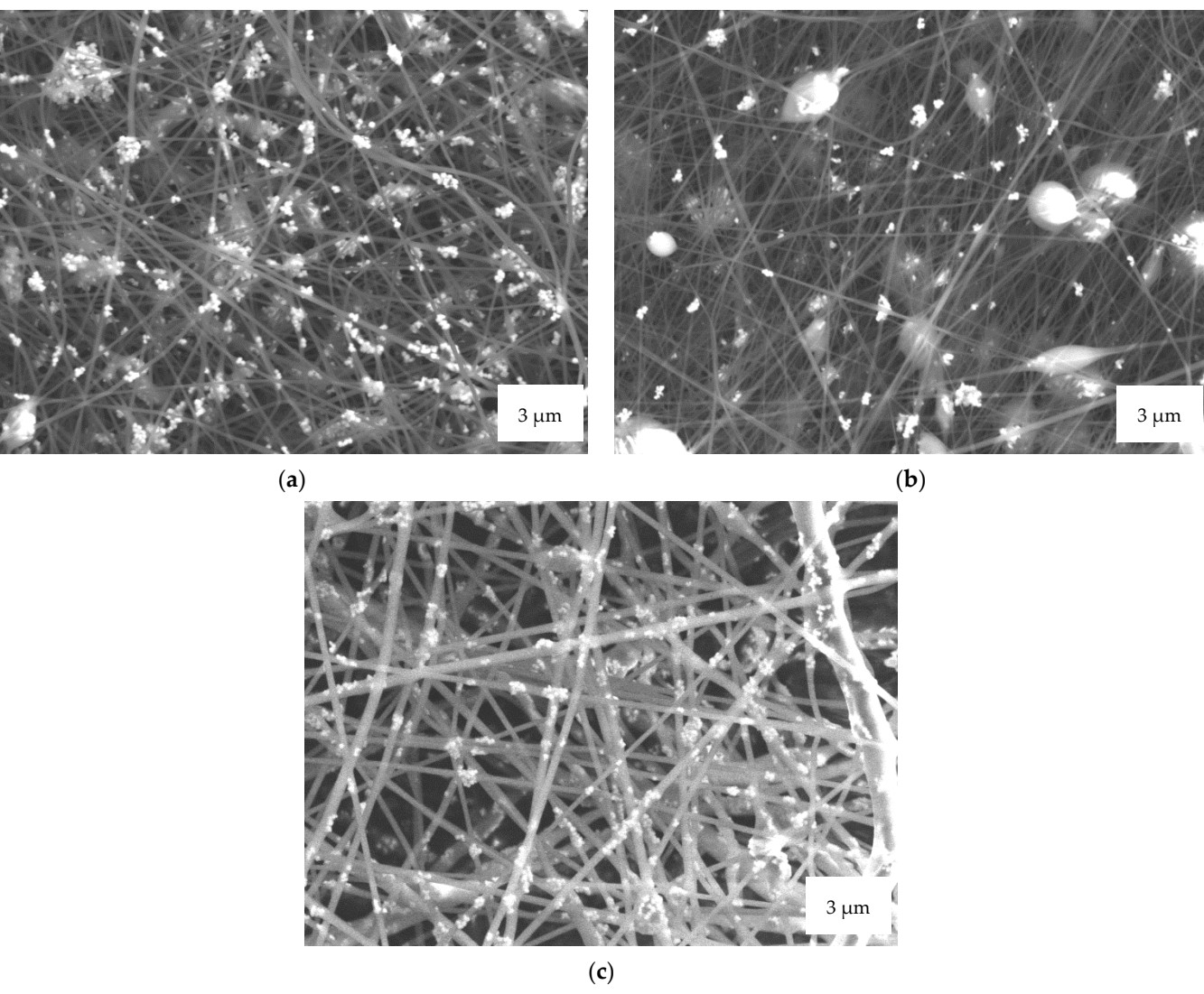

**Figure 6.** SEM micrographs of the magnetic nanofiber mats: (**a**) 16 wt% PAN with 65 kV voltage and 40 min ultrasonic bath; (**b**) 16wt% PAN with 80 kV voltage and 40 min ultrasonic bath; (**c**) 16 wt% PAN and 80 kV voltage without ultrasonic bath.

At 65 kV voltage (see Figure 6a), more agglomerations are detected and the beads are irregular in shape rather than spherical, while at 80 kV voltage (see Figure 6b), less agglomerations are detected and the beads have a more spherical shape. The nanofiber mat in Figure 6c shows agglomerations and the MNFs' diameter distribution appears to be significantly broader compared to nanofiber mats in Figure 6a,b. In addition, fewer beads are seen in Figure 6c than in Figure 6a,b, and the MNFs are not single and well-separated from each other as in 6a and 6b, instead, nanofibers are connected to each other.

The nanofiber diameter distributions are shown in Figure 7. The nanofiber mat produced from 16 wt% PAN at 65 kV voltage using an ultrasonic bath shows a diameter distribution of (159 ± 49) nm, and at 80 kV voltage using an ultrasonic bath presents a diameter distribution of (104 ± 45) nm, as well as (287 ± 82) nm for nanofibers fabricated at 80 kV voltage without an ultrasonic bath (see Figure 7a–c). However, the nanofiber mat prepared from 18 wt% PAN exhibits a diameter distribution of (247 ± 75) nm at 65 kV voltage using an ultrasonic bath, as well as a diameter distribution of (169 ± 59) nm at 80 kV voltage using an ultrasonic bath, and (289 ± 98) nm for nanofibers produced at 80 kV voltage with no ultrasonic bath. When comparing the nanofiber mats of 16 wt% PAN and 18 wt% PAN, it was found that the diameter of the nanofibers was larger in 18 wt% PAN compared to 16 wt%. This observation is known from previous test series and is due

to the increased polymer concentration in the electrospinning solution, which results in thicker nanofibers.

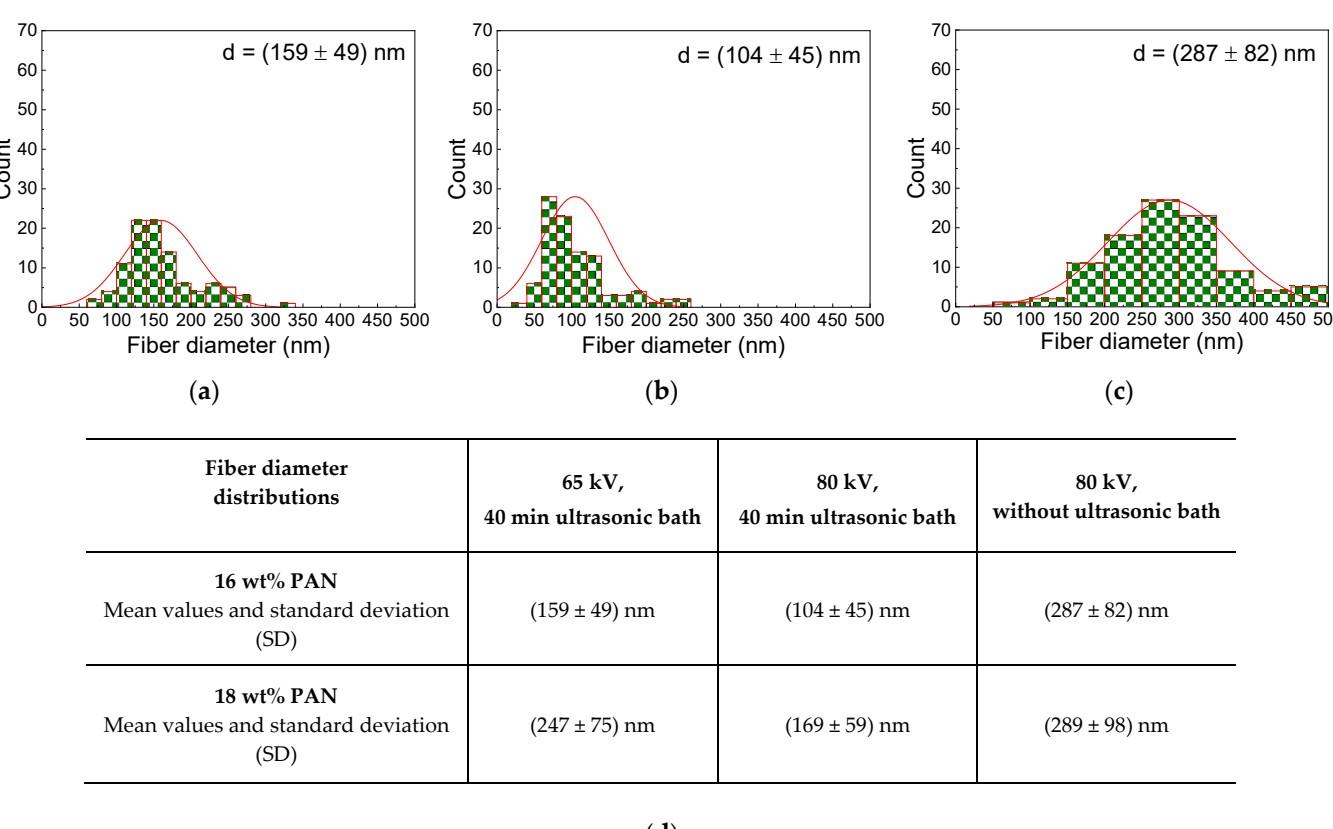

**Figure 7.** Diameter distributions of the magnetic nanofiber mats: (**a**) 16 wt% PAN with 65 kV voltage and 40 min ultrasonic bath; (**b**) 16 wt% PAN with 80 kV voltage and 40 min ultrasonic bath; (**c**) 16 wt% PAN and 80 kV voltage without ultrasonic bath; (**d**) Fiber diameter distributions of nanofiber mats from 16 wt% PAN and 18 wt% PAN.

Figure 8 gives an overview of the effects of the concentration of PAN and the ultrasonic bath as well as the applied voltage on the diameter distribution of the magnetic nanofiber mats. As suspected from the SEM micrographs in Figures 5 and 6, the nanofiber diameter distribution at 65 kV voltage (see Figure 7a) is broader compared to nanofiber mats produced at 80 kV voltage (see Figure 7b). Furthermore, without the use of ultrasonic bath (see Figure 7c), the nanofiber diameter distribution is significantly broader than with the use of ultrasonic bath (see Figure 7a,b).

In summary, it can be concluded that ultrasonic treatment of the electrospinning solution exerts an influence on the resulting nanofiber mat morphology and allows forming more uniform, well-separated and not interconnected nanofibers with smaller nanofiber diameter distribution and less agglomerations (see Figures 5a,b and 6a,b).

Besides the morphology, the distribution of nanoparticles also has a high importance on the defined applications. Therefore, Figure 8b,c show EDX maps of the magnetic samples. The investigated areas are marked with a blue frame (see Figure 8). EDX spectrum 1 defines a small area containing a bead (see Figure 9a,b), while EDX spectrum 2 was taken at nanofibers (see Figure 9a,c).

Looking at the EDX spectra 1 (Figure 8b), it is noticeable that the overlapping Fe L$\alpha$ and Fe L$\beta$-peaks at 0.705/0.718 keV in the EDS spectrum 1 taken on the bead are larger than in the EDX spectrum 2 correlated with the nanofibers (see Figure 8c). Moreover, the EDX spectrum 1 shows strong peaks of the element iron (Fe) and confirms the assumption that the magnetite nanoparticles are agglomerated within the beads.

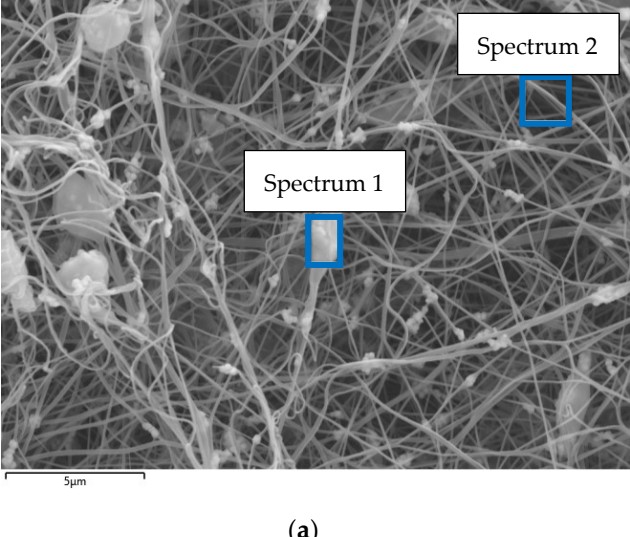

(**a**)

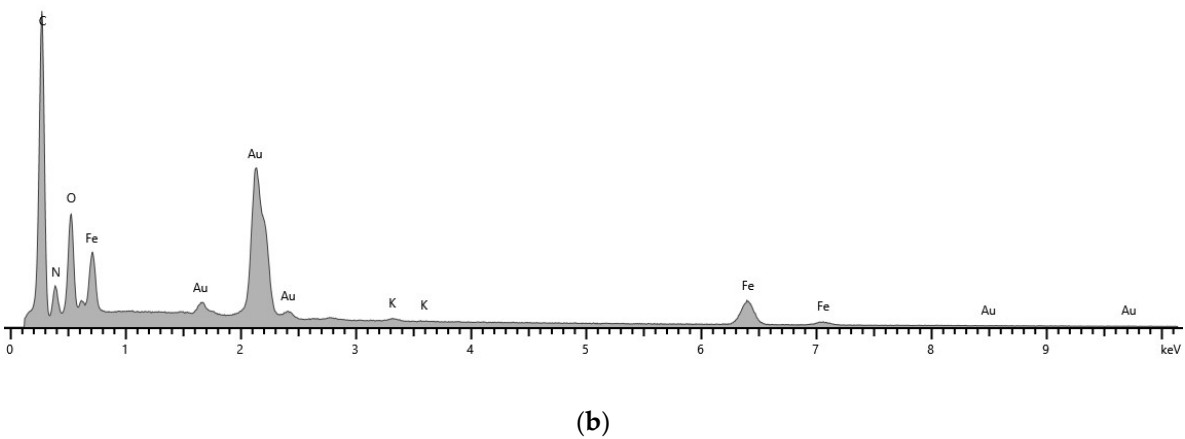

(**b**)

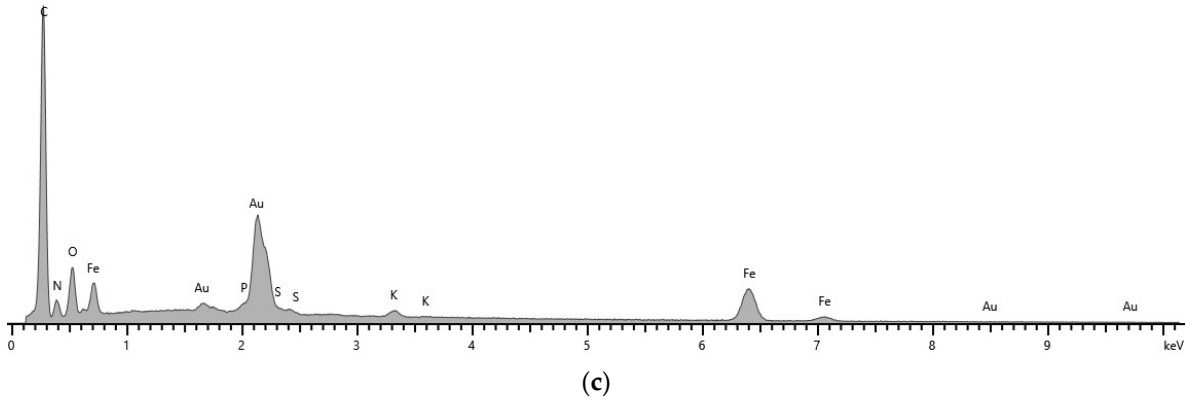

(**c**)

**Figure 8.** Energy dispersive X-ray spectroscopy (EDX) graphs of the magnetic nanofiber mats: (**a**) 16 wt% PAN with 65 kV voltage and 40 min ultrasonic bath; (**b**) Spectrum 1—spot on the bead; (**c**) Spectrum 2—spot on the nanofiber.

In the case of magnetic nanofibers, not only the surface morphology but also the distribution of magnetic particles in the nanofiber mats plays a major role. Figure 9 shows EDX maps of the magnetic samples. Figure 9a shows the iron indicative of the magnetite nanoparticles, and some larger agglomerations can be detected, as in our previous study [5]

where larger agglomerations were found in the spheres as well as smaller amounts of clusters in magnetic nanoparticles in the fibers. Figure 9c represents the amount of carbon indicative of the polymeric content of the nanofiber mats. In Figure 9b,c, some local variations in nanofiber structure as well as globules and agglomerations can be seen and magnetic particles appear to be well distributed. These observations differ from our previous study [35], where large concentrations of magnetic particles are hardly detectable.

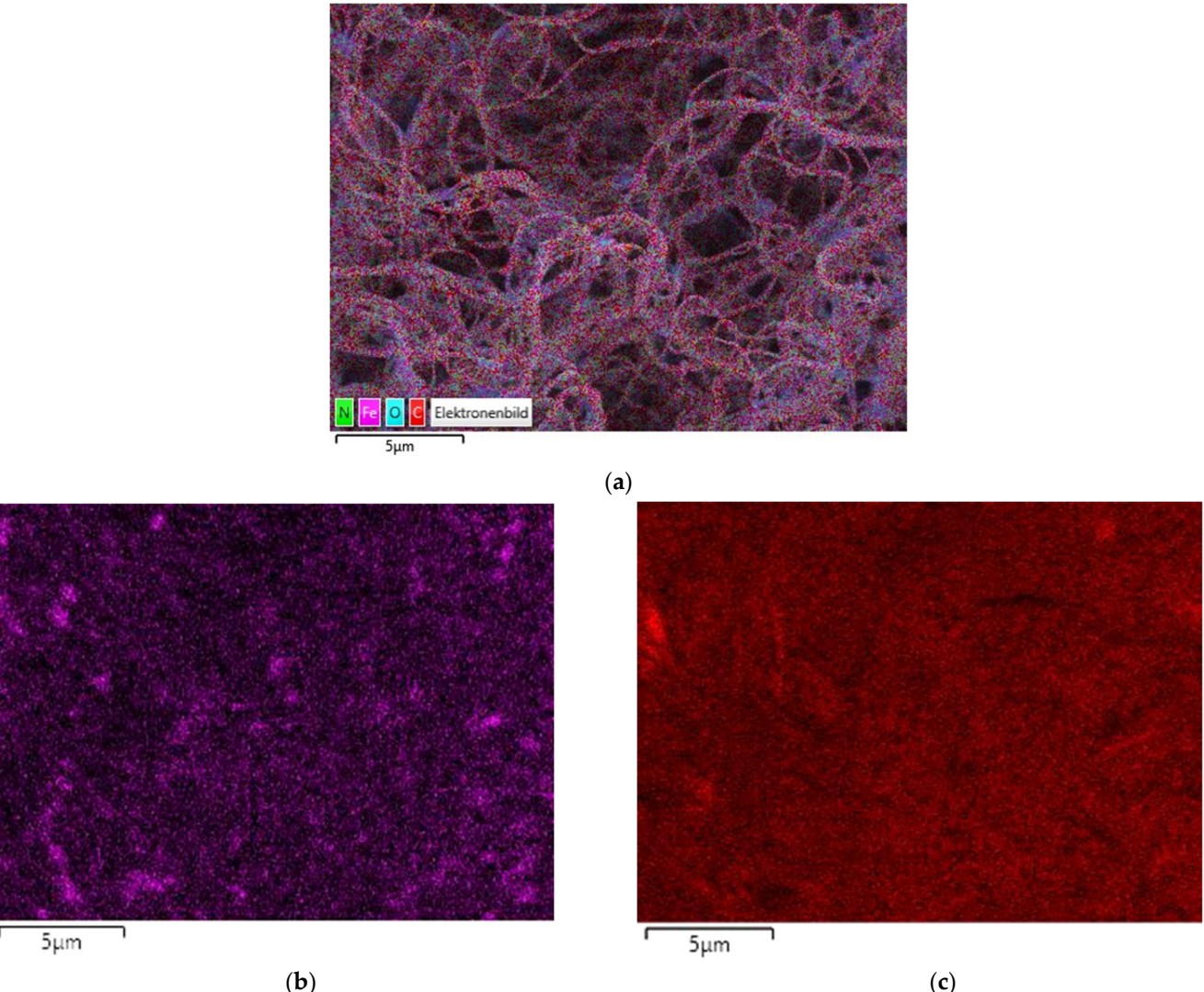

**Figure 9.** Energy dispersive X-ray spectroscopy (EDX) maps of the particles distribution of 16 wt% PAN magnetic nanofiber mats; (**a**) EDX map of Fe = iron, C = carbon, N = nitrogen, O = oxygen elements; (**b**) EDX map of Fe = iron; (**c**) EDX map of C = carbon.

Figure 10 shows characteristic FTIR spectra of PAN and PAN/magnetite nanofiber mats. Chemical investigation by FTIR shows the typical peaks of PAN, $CH_2$ bending and stretching vibrations at 2938 cm$^{-1}$, 1452 cm$^{-1}$ and 1380 cm$^{-1}$. In addition, the stretching vibrations of nitrile group at 2240 cm$^{-1}$ are visible as well as carbonyl stretching peak at 1731 cm$^{-1}$ [36]. In general, metals are known to cause deviations from a flat baseline [35]. The artifact at 2100 cm$^{-1}$ can be attributed to the incompletely compensated absorption of the diamond ATR crystal.

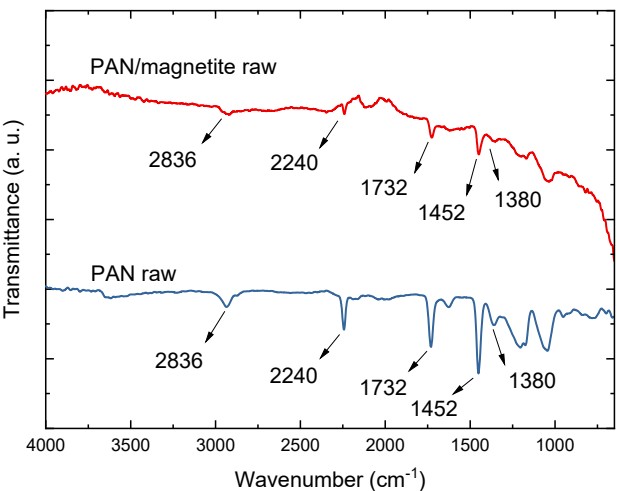

**Figure 10.** Fourier transform infrared (FTIR) spectra of exemplarily chosen nanofiber mats.

## 4. Conclusions

In this study, nanofiber mats were manufactured by simple addition of magnetic $Fe_3O_4$ particles to a polymer solution using needle-free electrospinning technique. According to this study, electrospinning parameters such as high voltage, polymer concentration, and the use of ultrasonic treatment exert an influence on the distribution of magnetic particles and fiber diameter distribution on resulting nanofiber mats. At 65 kV voltage, the nanofiber mats contain a broader fiber diameter distribution than nanofiber mats produced at 80 kV voltage. At 80 kV, without ultrasonic treatment, the nanofiber mats contain the broadest nanofiber diameter distribution. The effect of interfiber bonding and denser beads in Figure 5c should be investigated in further study to determine that settings used in this study result in the same nanofiber morphology, because it is of great importance to produce nanofiber mats with defined surface morphology for specified applications.

Moreover, agglomerations of magnetite nanoparticles were detected in the beads using SEM and EDX spectra. Similar to our previous studies [5,22,27], larger amounts of magnetite are detected in the beads than in the nanofibers.

Magnetic nanofibers are very promising for various applications. For example, beads of magnetic particles in the fiber matrix are important for data storage and neuromorphic computing, and regular magnetic nanofiber mats are suitable for energy storage applications.

**Author Contributions:** Conceptualization, L.S.; methodology, L.S. and A.M.; investigation, A.M., M.K. and L.S.; writing—original draft preparation, A.M. and L.S.; writing—review and editing, all authors; visualization, A.M., T.B. and L.S.; supervision, L.S. and T.B. All authors have read and agreed to the published version of the manuscript.

**Funding:** The authors acknowledge personal funding from the internal PhD funds of Bielefeld University of Applied Sciences.

**Institutional Review Board Statement:** Not applicable.

**Data Availability Statement:** The data created in this study are fully depicted in the article.

**Conflicts of Interest:** The authors declare no conflict of interest.

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
