# Peer review of "Investigation of the Morphological Structure of Needle-Free Electrospun Magnetic Nanofiber Mats"

_magnetochemistry, doi:10.3390/magnetochemistry8020025_

Round 1

Reviewer 1 Report

This work investigated the morphological structures of the magnetic nanofiber mats under different applied voltages and ultrasonic conditions. Both SEM images and FTIR spectra were employed to study their structural changes. This work can be published on the Magnetochemistry after addressing the following questions.

  1. Are the magnetite nanoparticles stable in the electronspun mats? Will they be easily washed away from the polymer network?
  2. Clear agglomeration of the magnetite nanoparticles has been seen in all the samples. Does this because those particles were not dispersed well before the electrospinning? I noticed that authors mentioned that they only mixed the particles and PAN solutions by manual stirring, which is generally not enough for a good mixing.
  3. It is confusing that authors mentioned that the mats in this work were prepared via needle-free electrospinning method. But in the method part, a nozzle size of 0.9mm was reported. It would be great authors can elaborate a little bit more on the information of the setup used in this work.
  4. Scale bars of the SEM images are too small to identify. Suggest increasing the font size. It is also suggested to keep the style of scale bars the same.

Reviewer 2 Report

In the abstract, add a space “ 20wt%.”

At the end of the introduction section, what is new in this study? Can the authors highlight the main goal or hypothesis that led to this work?

For the materials and methods:

  • Was the magnetite added in a powder form? Was there any coating on the magnetite surface? If not, what was the surface charge?

  • please provide the ultrasonic bath type/model.

  • Voltage values ranging from 65 kV to 80 kV seem high. What happens if the electrospinning process is performed at other ranges? What is the reasoning for selecting these high voltages? Page 3, line 123. The word cf. is not needed.

Page 4, line 132. Unclear what “membrane areas” mean?

Page 4. The authors mentioned: “The individual MNFs look significantly thicker (Fig. 3c) 136 compared to nanofiber mats in Fig. 3a and Fig. 3b”. Please provide statistical analysis to make this claim. Also what is the difference in the diameter? The same applies to Figs. 4a and 4b (page 5, lines 142 – 145). The authors should make a table contrasting the diameters obtained.

Page 7. It seems the particles are randomly agglomerate throughout the fibers producing beads. Inter-fiber bonding also is observed in fig 5c. The authors should discuss this phenomenon and explain if it is reproducible or not.

Page 8. The SEM images confirm the random distribution and aggregation of the particles. Huge nodules containing magnetic clusters are seen. 

Page 9. Should be well-separated instead of “well-separeted”

Page 9. The peak intensity for Fe is relative and not directly comparable. Moreover, spectrum 2 collects scans, including the hollow background. To honestly assess the presence of nanoparticles within the fibers, a cross-sectional cut and TEM imaging is recommended. See for instance:

https://doi.org/10.1021/acs.langmuir.8b01167

Page 10. The authors should expand this discussion: “It is known that metals generally cause deviations from a flat baseline.” Also, is there a Fe-O peak in the FT-IR that can be discussed?

Additional weaknesses:

-            The authors should explain more in-depth the ultrasonic treatment. It is suggested that the particles are sonicated and dispersed in the DMSO solvent first and then mixed with the PAN solution. Otherwise, it is unclear if magnetite achieves proper dispersion. DLS could confirm the particles' aggregation level within DMSO or the DMSO/PAN solution.

-            The nanoparticles are aggregated, and particle distribution seems impossible to control.

-            What are the magnetic properties of the beads? And the final percent of magnetite embedded within the fiber. Thermogravimetric analysis is highly recommended.

-            What is the stability of the Fe3O4? XRD should confirm the crystallinity of the sample before and after treatment.

Round 2

Reviewer 2 Report

All comments to my initial reviews were addressed. The study can help future researchers in this challenging subject.